# Healthcare workers' beliefs, practices and experiences regarding the COVID-19 pandemic and resulting governance from preparedness and response plans in Faranah, Guinea

Carlos Rocha[1]*, Moussa Douno[2,3], Lena Landsmann[1], Kamis Diallo[4], Rebekah Wood[1], Mardjan Arvand[1], Carolin Meinus[1], Mamadou Diallo[4], Almudena Marí-Sáez[1,5]

1 Robert Koch Institute, Berlin, Germany, 2 Centre d'Excellence Africain pour la Prévention et le Contrôle des Maladies Transmissibles, Université Gamal Abdel Nasser de Conakry, Conakry, Guinea, 3 Centre National de Formation et de Recherche en Santé Rurale de Maférinyah, Forécariah, Guinea, 4 Faranah Regional Hospital, Faranah, Guinea 5 Institut de Recherche pour le Développement, IRD_UMI 233 TransVIHMI Délégation Régionale Occitanie, Montpellier, France

* rochac@rki.de

## Abstract

There is a growing interest in the implementation of people-centered components within outbreak preparedness and response plans and their resulting governance. Information on this topic still remains scarce, particularly in settings such as Guinea in West Africa. In the context of COVID-19, we conducted 21 qualitative interviews with healthcare workers (HCWs) from the Faranah Regional Hospital in Guinea to explore pandemic preparedness and response adherence. We also analyzed corresponding senses of vulnerability and risk, information sources, perceived information reliability, associated rumors and previous outbreak experiences. To achieve a broader analysis, we employed the "therapeutic landscape" concept, which conceives health provision as a process constructed through practices, beliefs and experiences. A recurring theme across the interviews was the abundance of information, which caused confusion mainly due to rumors and a lack of trust in the country's health system. Initial fear gradually diminished due to the disease's low mortality rate in the country. HCWs described preparedness enforcement in the city of Faranah that started strongly but gradually relaxed over time. This waning of enforcement marked a difference between the city and the hospital. HCWs appraised the first positive case declaration in the city to be the most significant moment, followed by a strong community reaction hindering the pandemic governance. We conclude that preparedness and response plans must better situate their interventions socially and devote more structural efforts to incorporating the social landscapes of diseases and outbreaks. Incorporating these social landscapes facilitates an understanding of the operational barriers to people-centered approaches. It also serves as an indicator for strengthening infodemic detection and management. Responses should therefore consider 1)

**Data availability statement:** We are not able to make our underlying data set publicly available for ethical and data protection reasons. The data contain potentially identifying information: our data have been collected from a small group of participants, and even data that are not directly identifying can become identifying in combination (e.g. sex, profession, department ward, work description). This restriction to data availability has been imposed by the Robert Koch Institute's Data Protection Office. Data requests may be sent to the ZIG Department Secretariat via ZIG-Assistenz@rki.de.

**Funding:** The study was conducted as part of the PASQUALE project (Partnership to Improve Patient Safety and Quality of Care), a partnership between the Robert Koch Institute, the FRH and Bouaké University Hospital (CHU) in Ivory Coast and funded by the Global Health Protection Program (GHPP) (Grant code: ZMVI1-2519GHP710). Due to the COVID-19 pandemic, funder allocated project resources for COVID-19 related capacity-building activities and research, from which the data of this paper stems. The funders had no role in study design, data collection and analysis, decision to publish, or preparation of the manuscript.

**Competing interests:** The authors have declared that no competing interests exist.

**Abbreviations:** ANSS, Agence Nationale de Sécurité Sanitaire; CT-EPI, Center for the Treatment of Epidemics; EVD, Ebola Virus Disease; FRH, Faranah Regional Hospital; HCW, Healthcare workers; IDI, Semi-structured interview.

hospitals applying different pandemic understandings that transcend biomedical and scientific orders, and 2) HCWs as portraying shifting pre-existing identities leading to marked in-/out-group distinctions, which directly influence healthcare, risk perception, and information and rumor management.

## Introduction

In the aftermath of the 2014 – 2016 Ebola Virus Disease (EVD) outbreak in West Africa, the social sciences have become increasingly involved in outbreak preparedness and response, both in Africa and worldwide [1 –4]. The importance of a context-integrated approach to prevention, preparedness and response was further highlighted by the subsequent COVID-19 pandemic [5]. Context-specific and community-centered approaches are considered key for comprehending local understandings of diseases, identifying multisectoral drivers of vulnerability, strengthening local resilience networks, and developing effective community engagement models in the context of both the COVID-19 pandemic and future outbreaks [6–8].

International and national preparedness and response plans for the African continent emphasized the need to engage communities around COVID-19 by reinforcing the capacity of local and community leaders, civil society groups and healthcare workers (HCWs) [9]. Despite this growing awareness, information on the implementation and governance consequences of context-specific and community centered approaches still remains scarce [10,11].

While disastrous effects related to the COVID-19 pandemic were initially expected in Sub-Saharan Africa due to its fragile healthcare systems, the region recorded lower infection and mortality rates than most countries in Europe, America and Asia [12–14]. Nonetheless, the impact on health and economic systems, as well as on livelihoods and citizen-state relationships, was considerable [15]. This highlights the need for research to shed light on these structural tensions and their impact on national and international preparedness and response plans [16,17].

At the beginning of the pandemic in Guinea, which was marked by its first case in March 2020 [18], the National Preparedness Plan included disseminating appropriate information on the disease, its mode of transmission and prevention. The National Response Plan aimed to increase capacity and communication through raising awareness, radio and television broadcasting, producing communication tools, and community engagement [19]. While the response strategies recruited more HCWs, Infection Prevention and Control (IPC) training on COVID-19 was lacking or unsatisfactory [20]. Rapid situational assessments among the country's HCWs showed a marked disparity in epidemiological knowledge between the capital (Conakry) and the interior regions, where knowledge levels were lower [21,22]. Social media and traditional media (radio and television) were identified as the primary source of information for HCWs in both rural and urban settings [21,22].

In a post-Ebola context where the population is highly suspicious of the health system [23–25], the arrival of the COVID-19 pandemic may have exacerbated existing negative perceptions of HCWs (e.g., perceived as potential sources of contagion)

[26,27]. This new pandemic may also have contributed to maintaining distrust and further decreasing the use of health services, thereby exacerbating inequitable patient care and hospitalization conditions [28]. Nonetheless, the country, particularly its peri-urban and rural areas, reported relatively fewer positive cases of COVID-19, as well as lower rates of morbidity and mortality compared to global rates [28,29].

HCWs are at the frontline of the response to infectious disease outbreaks. Low perceived preparedness for the response to the COVID-19 pandemic was associated with stress and burnout due to mental strain, physical exhaustion and separation from families and loved ones [30–32]. Structural challenges in the health system, such as inadequate infrastructure, a lack of universal coverage and poor working conditions, created an unhealthy and corrupt therapeutic environment for HCWs [33–36]. These factors increase the risk of healthcare facilities becoming hotspots for outbreaks or mistrust in pandemic conditions [37].

More research is needed to provide evidence on the implementation of local response and preparedness plans. A comprehensive understanding of health facilities is necessary to contextualize hospitals and their practices as products of social, political, and biomedical factors that influence disease transmission [38–40]. This in-depth understanding enables to complexify HCWs' experiences and the roles expected of them by preparedness and response plans [41]. In order to substantiate the role of healthcare workers (HCWs) in responding to the pandemic, we employ the concept of the "therapeutic landscape". We approach health provision as a lived, understood and constructed process through the actions of a group of people in a particular place [42–44].

This study explores some of the emergent features of the therapeutic landscape at the Faranah Regional Hospital by analyzing the beliefs, practices and experiences of its HCWs with regard to preparedness and the response for COVID-19 at the hospital and in the city of Faranah.

## Materials and methods

### Ethics approval

Ethical approval (approval number: 036/CNERS/21) for the study was granted by the National Health Research Ethics Committee of Guinea (*Comité National d'Éthique pour la Recherche en Santé).*

### Study design and study setting

From 1st August to 31st October 2021, we conducted a three-month qualitative study based on a grounded theory approach. This involved an iterative process of data collection and analysis, and comprised semi-structured interviews (IDIs) with healthcare workers (HCWs) at the Faranah Regional Hospital. The Faranah Regional Hospital is a public hospital in the Upper Guinea Region. It employs 74 HCWs and serves as a referral hospital for the Faranah region, covering a population of more than 300.000 which belongs mainly to the *Malinké* ethnic group. The hospital consists of 16 wards, including a surgery, laboratory, maternity ward and a Center for the Treatment of Epidemics (CT-EpI).

### Participant selection

All HCWs (medical doctors, nurses, midwives, nursing assistants, laboratory technicians, engineers and biologists) were eligible to participate in the study. The main inclusion criteria were being a HCW in one of the hospital's wards and willingness to participate in the study. We initially used convenience sampling aiming at a representative distribution of age, professional groups and between genders. Following a snowball methodology, each participant was asked to recommend a colleague who could be interested in participating in the study. While convenience sampling might have some limitations as opposed to purposive sampling in terms of the potential lack of accurate representation of the target population and limited generalizability, we employed the following strategies to minimize the shortcomings. The snowball methodology and the iteration between data collection and analysis contributed to achieving a more

representative distribution across age, profession and gender. This also allowed us to broaden our sample in order to ensure the representation of multiple perspectives and ensured access to participants who might not have been captured using purposive sampling alone.

## Data collection

Written informed consent in French was obtained from all study participants. The study had previously been approved by the hospital management. Before starting data collection, the researcher responsible for data collection (DM) introduced the purposes and objectives of the study to the hospital's HCWs during a staff meeting. With the HCWs who consented to participate, a meeting was scheduled for a day, time and place that made them most comfortable. Data was collected at the Faranah Regional Hospital. Each interview session lasted between 22 and 75 minutes. The interviewer (MD) was already known to the participants from previous research carried out at the hospital, ensuring a familiarity with the data collection context and a trust rapport with the hospital's HCWs [45]. Interviews were held mainly in French and/or Maninkakan a local language familiar to the interviewee and the interviewer. MD translated the data if needed and transcribed verbatim. MD and CR analyzed the data in French. Quotations were translated into English for the purposes of this paper.

## Interview guide

We developed a semi-structured interview guide based on the social sciences literature on previous epidemics in the country and on the COVID-19 literature available at the moment. Questions included perceptions about the pandemic progression over time, the sources and changes in the type of information about it as well as associated memories and emotional appraisals. The guide also covered opinions and mental images about the virus and the pandemic as well as questions about the perception of the changes caused by the enforcement of preparedness and response strategies in the hospital, the Faranah Region and the country.

## Data analysis

We performed a preliminary iterative analysis ensuring data collection and analysis followed by an inductive coding on a random selection of transcripts. This resulted in the creation of a codebook which was then applied to all of the data using NVivo 12 qualitative data analysis software. Following a grounded theory approach, we carried out a subsequent inductive analysis to assess the inclusion of new theoretical frameworks for a deeper social understanding of the local uptake of outbreak preparedness and response plans. The theoretical framework of hospitals as therapeutic landscapes was identified as the most appropriate as it has been applied in similar contexts to analyze health care provision [43]. It views hospitals as both material and symbolic structures creating and created by social forces, some anchored in historical processes like the bureaucratic organization of healthcare provision from the colonial to the post-independence administration [46,47]. Furthermore, the concept acknowledges that healthcare provision acquires social meaning through the beliefs, practices and experiences of the people that practice it in a particular place [44]. The HCWs' identity incorporates the clinical/medical practice (e.g., profession or years of experience) and community social leadership as well as worldviews (e.g., magical or religious) that are intrinsic to health care provision [37]. Data was finally deductively analyzed following the main three categories within the therapeutic landscape framework: beliefs, practices and experiences. In relation to health, we define a belief as a precept accepted as true or real attached to illness and cure [48]; a practice as the application of an idea or a belief by the actions of social actors within given societies [48] and an experience as a practical contact with and observations of facts and events [49]. Further information concerning the data analysis process can be found in the supporting information table (S1 Table).

### Techniques to enhance trustworthiness

To enable the correct development of the interviews, MD took field notes, which were shared with the other team members (CR and AM-S), to provide timely feedback, ensure the iteration of data collection and analysis, and guarantee the quality of the data until data saturation was achieved. This occurred when no new information emerged in any of the interview guide topics.

## Results

We interviewed 21 HCWs from different hospital services (general medicine, surgery, radiology, maternity, CT-EpI, among others) and professions. We outline participant characteristics in Table 1.

We present the results under the following three categories dedicated to beliefs, practices, and experiences.

### HCWs' beliefs about COVID-19

This section presents types and reliability of information sources, as well as perceptions about rumors and risk.

### Information content and sources

Participants explained receiving information about the appearance of a new and highly contagious virus responsible for a severe respiratory syndrome originating in China. The information included topics like the nature of the virus, virulence, its

**Table 1. Participants' characteristics (N = 21).**

| Characteristics: | N = 21 (%) |
| --- | --- |
| Gender: | |
| Female: | 11 (52%) |
| Male: | 10 (48%) |
| Age: | |
| 20 – 29 | 4 (19.0%) |
| 30 – 39 | 9 (42.9%) |
| 40 – 49 | 2 (9.5%) |
| 50 – 59 | 5 (23.8%) |
| 60+ | 1 (4.8%) |
| profession: | |
| physician | 5 (23.8%) |
| Nurse | 6 (28.6%) |
| Midwife | 2 (9.5%) |
| Laboratory Technician | 2 (9.5%) |
| Other | 6 (28.6%) |
| Services: | |
| Maternity: | 2 (9.5%) |
| CT-EpI: | 2 (9.5%) |
| General Medicine: | 2 (9.5%) |
| Emergency: | 2 (9.5%) |
| Pediatrics: | 2 (9.5%) |
| Surgery: | 3 (14.4%) |
| Laboratory: | 2 (9.5%) |
| Others (radiology, ophthalmology, acupuncture, dental cabinet, pharmacy and maintenance) | 6 (28.6%) |

prevention and control. One theme that HCWs consistently articulated was the case fatality rate, which they qualified as high and thus unsettling. The primary sources of information were the media. Overall, participants mentioned social media (mainly Facebook), television, radio and the Google search engine. In addition, a few participants explained getting further information at the Faranah Regional Hospital through the director and department heads during staff meetings.

### Reliability of information sources

While some of the interviewed HCWs qualify radio and television as reliable to learn about the pandemic, others found that the most credible information sources were the websites from, e.g., the WHO, UNICEF or OIM, the communiqués from the Ministry of Health or the National Agency for Sanitary Security (ANSS) and information from the hospital directives. One participant summarized this opinion by stressing the reliability of leadership figures:

'The good information is obtained through […] doctors, it is the doctor who must give the right information so that we can better avoid this disease and raise awareness to improve the health of the population' (IDI 19).

Some argued that the most trustworthy source of information was the Guinean state television that broadcasted ANSS "communiqués" every evening. However, the latter was questioned by others since the community death counting was perceived as flawed. Because of this, one HCW reported that:

'The international channels are especially reliable, because […] they tell the exact figure per day, the number of deaths, the number admitted in resuscitation (ICU)' (IDI 14).

For these reasons, the international media elicited credibility, considering that 'the statistics they put out are not biased' (IDI 14) since they report the exact numbers of cases and deaths.

### Conflicting and reassuring information

Contradictions about tests results were especially prominent when enquired about conflicting information. As one HCW said:

'Sometimes we tell them today [to the patients], you're positive, after a few hours, we call them and tell them, you're negative, so that's a contradiction' (IDI 4).

HCWs argued that this is a source of considerable confusion both for them and patients since it lead to questioning the reliability of antigen rapid tests, which they described as being sensitive to many other viruses.

Another participant explained that many patients underestimated the danger of the disease by equating it with influenza or malaria. In this sense, they claimed that it already existed in the country or that the Guinean population had antibodies. One HCW explained:

'For example in Conakry, it is said that the disease is not true, others even say that it is "corona-palu" [alluding to the word 'paludisme', another denomination for Malaria in francophone West Africa]' (IDI 6).

All participants stressed that there was also reassuring information. For the interviewed HCWs the most comforting was the high recovery rate among admitted patients in Guinea. They also reported that low case fatality rates gave them hope that a treatment would eventually exist. In addition, they found the information on preventive measures as reassuring because:

'When they are respected in the services and the families, there will be no contamination. What is also reassuring for me is when, out of 100 sick people, we say that 80 to 90 have been cured, or out of 100 people, there have only been 2 deaths, that is really reassuring' (IDI 9).

They also perceived information on the development of vaccines as encouraging.

Other participants felt reassured with the pandemic's progression in Europe and North America:

"Because Africa is a developing continent, some people think that [epidemics] are only for Africans, because we have nothing. When I find that the same problems exist in highly developed countries, people are dying, it allows me to conclude that it is a global problem" (IDI 13).

### Risk perceptions

The risk perceptions presented by the HCWs were sometimes in contradiction with the information gained from the biomedical practice (detailed above). One theme that participants consistently articulated was the perception of high risk brought by this disease. The worldwide high mortality rate, the contagiousness and the economic and socio-cultural consequences were held accountable for this elevated risk perception. They also associated this with the fact that the pandemic was a significant problem worldwide. As one HCW put it:

'From the moment I saw [information on COVID-19 on] these continents [i.e., Europe and North America], I kept seeing the number of corpses, the number of deaths, and I was afraid, I said, listen, if the people we rely on are in trouble like this, if it happens in Africa, what will happen?' (IDI 10).

Fear and sadness were the most prominent emotions evoked about the received information. These emotions were elicited by the awareness that this disease had foreign origins and was being brought to the country by people able to travel. One HCW explained:

'I was really scared. Especially of the people who have traveled outside of the country. Even my son is abroad, he even wanted to come […] but as soon as the problem started, I talked with him and I told him not to come, to stay there' (IDI 3).

Furthermore, as one participant put it:

'Once you are told that the ministers are attacked, the senior staff is attacked, you must be afraid since the disease has not spared the authorities' (IDI 12).

Participants described a change in the perception as the COVID-19 pandemic evolved. Several described how their initial fear gradually diminished mainly due to the low mortality rate in Guinea, the decreasing number of positive cases, and the high recovery rate among hospitalized patients.

### Rumors about COVID-19

Overall, participants reported hearing community members denying both the presence of the virus in Guinea and the emergence of a new, deadly disease. Participants noted that various rumors and conspiracy theories circulated about this new infection. According to them, rumor sources included "word of mouth," the radio, print and social media. Participants consistently linked the rise of these rumors and conspiracy theories with the country's political situation as the pandemic coincided with 2020 presidential elections. The president in office sought, with success, to run for a third term.

Almost all participants perceived these rumors and conspiracy theories as untruthful. Yet, they did not report any efforts by the authorities or the Faranah Regional Hospital to debunk them. HCWs explained that the high level of illiteracy in the Guinean population facilitated their rapid spread. One participant argued:

'The high amount of contradictory information is quite normal, living in an environment where the illiteracy rate is so high. 80% of the population says that the disease doesn't exist, that it's invented' (IDI 10).

Other participants emphasized social and religious factors such as inter-ethnic tensions. One HCW understood the emergence of this disease as a punishment from God because of humanity's trespasses:

"Such a disease came. Is it not a sanction of God? [...] in Italy they allowed in a church the homosexual marriage. God doesn't like that" (IDI 7).

### Practices related to COVID-19

This section presents perceptions about the enforcement of response plans and their implementation as well as strategies to counteract vulnerability and rumors.

### Counteracting vulnerability and rumors

All the participants reported that the means of feeling protected against the pandemic were the infection prevention and control measures, especially hand hygiene, wearing masks and complying with social distancing. HCWs estimated that community engagement was effective in counteracting rumors and enforcing IPC.

All HCWs reported having heard of non-biomedical treatments for this new infection, as well as medications designed to treat other infectious diseases. They mostly heard about herbal medications, including artemisia, cinchona bark, lemon, garlic, acacia, ginger, and other local plants they named in the local language: "Popa" and "Sindjan lilin". Some participants also mentioned that the idea that the virus was susceptible to bitter plants was widespread. This motivated people to drink the cinchona-based decoctions and/or "Popa." While some participants said they had seen acquaintances using some of these plants, only one HCW claimed to have bought, prepared and drunk one with their children at home. This participant acknowledged: 'I advised people, even my children who are in Conakry, to boil and drink it' (IDI 7).

Most of the HCWs stated they did not believe in the decoctions and other non-biomedical treatments. One HCW explained: 'these treatments I can say no, personally, as a healthcare worker, I do not prescribe them' (IDI 4). However, some were reluctant to admit whether or not they believed in these non-medical treatments, stating that what was said should not be ignored, given the absence of a treatment or vaccine at the time. One HCW explained:

'I respect what people say, we should not say that it's not true. Because we do not have a classic treatment [biomedical treatment] we are told to prescribe Chloroquine. For Westerners, we have to give Azithromycin, Paracetamol against fever, headaches and so on [...] You accept what people say and then you advise preventive measures until we have a vaccine' (IDI 20).

In addition, one participant mentioned prayer as means of protection against this disease: 'we must believe in God and pray that the disease does not attack us' (IDI 7).

### Preparedness in Faranah and at the Faranah Regional Hospital

All the participants in this study confirmed that there had been a change in lifestyle in Faranah following the declaration of the country's first COVID-19 case. This change centered on the enforcement of the preventive measures, the closure

of places worship and leisure, and the prohibition of social gatherings. For one HCW, these restrictive measures were communicated:

'Through the media, radio, and even awareness-raising campaigns. Groups of people went from neighborhood to neighborhood to sensitize people on how to wear masks, even distributing handwashing kits and masks' (IDI 1).

Participants reported observing some changes at the Faranah Regional Hospital. They reported that a triage system and a handwashing station had been set up at the hospital's main entrance and in the wards. Some days later, security guards were hired to enforce preventive measures, such as mask-wearing and hand hygiene for those entering the hospital grounds.

Another change was a decrease in the number of HCWs in each service,particularly those without contract with the Guinean State, as well as trainees' ceasing to work to avoid overcrowding. Participants also described a decrease in the patients' attendance rates. People sought healthcare via pharmacies or non-biomedical treatments. This resulted in a perceived increase in the hospital's mortality rate as patients were admitted late with complications. Participants emphasized that the slowdown in hospital activity had an economic impact on them, since HCWs receive a share of the hospital's revenues from patient admissions and treatments at the end of each month.

Concerning triage and treatment, HCWs consistently articulated that there was confusion about suspect case definitions. Some claimed not receiving training about this new virus and treatment, while only a few mentioned both the case definition and treatment. Since the symptomatology resembled other diseases endemic to the region, one HCW reported:

'In our country, often, the diseases have common signs: it is fever, headache. Now, sore throat and cough are more specific for COVID-19. That is, in addition to all the malaria signs that we usually know, if there is cough and sore throat, and if the patient has lived in an area or has been in contact with a COVID-19 case, it is a suspect case' (IDI 20).

**Perception of the preparedness in and outside of the hospital**

Most participants reported relying on the preventive measures enforced by the hospital and the city. However, they did not feel entirely safe due to patients' lack of compliance. Many participants described a significant contrast between the city and the hospital:

'The difference is that in the hospital, we wear gloves and masks, we wash our hands, we wear the complete PPE for example in the lab; but in the city, we don't wear the gloves, we don't wear the gown, we only wear masks and wash our hands' (IDI 1).

This difference created a certain distrust between HCWs, patients and community members, since the former only felt protected and able to comply with preventive measures within hospital facilities. As another HCW argued:

'In the hospital, all the measures are there, the distances are respected, but in the city, you can't, for example we women who attend the markets. When we go to the market with the masks, some people look at us badly and you can't keep the distance' (IDI 10).

Other participants reported that the city's population started to neglect preventive measures because there had been no new recorded cases in the city. Additionally, the Guinean government's perceived failure to enforce containment measures in due time and utilize the most appropriate communication channels was considered conspicuous. One participant explained:

'In the community, practically, we must refer to the community leaders [for risk communication and community engagement]: people who are loved, who are listened to, who have the confidence of the population, are the ones who should give them this knowledge. [...]. Educational leaders [information multipliers] used to come from Conakry, they have the tenure, they go to the prefectures, that is *mamaya* [nonsense] [...] We have to go to the communities and choose the key persons there and train them and they will explain to their relatives [...] When we do that, we will have good results. Not everyone watches TV. In the villages, people don't have a TV. The radio, not everyone listens to' (IDI 5).

Participants who had been in contact with cases at the hospital reported that their colleagues from other services started to mistrust and avoid them for fear of becoming infected. One staff member described this as peers' stigmatization.

## Experiencing COVID-19

This section describes cultural and social reactions and the parallels established with the 2014 – 2016 EVD outbreak.

### Pandemic response in Faranah

HCWs appraised the declaration of the first positive case in the city of Faranah as the most significant moment. One HCW described:

'The moment that struck me the most was when a patient left another country for Conakry; there, his test was done but he fled to come to Faranah. This panicked the entire population' (IDI 9).

The participants experienced a community reaction in an attempt to counteract the spread of the virus. As one participant put it:

'In Faranah […] all households had hand washing stations and everyone wore masks. In addition to that, there was the military who monitored the community itself. Everyone was involved so that either you wear your mask because you want to avoid the disease or you wear it because you want to avoid getting fined' (IDI 4).

Other participants reported a mistrust among citizens, who avoided getting close to each other for fear of becoming infected. One HCWs explained:

'Yes, there were changes, there was a crisis of confidence between people. Even in vehicles, people did not approach each other and mosques were closed. It altered a lot the social relationships, because people were afraid [...] people were running away from each other, the contacts decreased a lot' (IDI 20).

### Community reactions

Following the confirmation of a positive case in the city, participants reported that the first community reaction was disease denial. Participants explained that there was subsequently an attempt to protest against HCWs, who were accused of falsely diagnosing their relatives as positive cases. One of the participants explained:

'When the first case was declared, the population had a negative reaction. First, they hid the person, they didn't give clear information and they said that because he came from the whites, they [the authorities] thought he had money, that's why they are looking for him, because he came from Spain. And even with the follow-up team, there were stones thrown at them in the neighborhood' (IDI 5).

 

Participants explained that the involvement of the town mayor was necessary. Together with the health authorities, he used the local rural radio to inform the population and reduce tensions. Other participants argued that the population was disappointed because HCWs could not find a treatment for this new infection.

HCWs explained that these tensions had led to a crisis of confidence between them and the population, with the latter believing the former to be responsible for spreading the virus. This resulted in mistrust, fear, and people avoiding the hospital. One participant pointed out that some people were afraid of the thermo-flash thermometer because they perceived it to detect positive cases. Another participant said that CT-EpI facilities prevented other people from going to the hospital. They feared that the infection would spread from the patients in this service.

## The shadow of EVD

One prominent theme that emerged was the 2014 – 2016 EVD epidemic. Participants drew many parallels to describe the COVID-19 outbreak. Experience acquired during EVD was reassuring factor in the response to this new pandemic. Nonetheless, this reassurance was somewhat undermined by the fact that the institutional structures and preparedness measures created in the aftermath of the EVD epidemic did not prevent the spread of the COVID-19 infection beyond Conakry.

Participants reported that the similarities between the EVD epidemic and the COVID-19 pandemic were clinical symptoms such as fever and physical asthenia, and preventive measures such as hand hygiene. One participant emphasized the infectious nature of both diseases, while another mentioned the population's mistrust of hospitals during both outbreaks. The participants relayed several differences between the two diseases. They mentioned new preventive measures introduced during the COVID-19 pandemic, namely mask wearing. In addition, the participants alluded to the modes of contamination and the significance of these diseases by mentioning the mortality rates.

Despite the high contagiousness of COVID-19 (referring to its rapid spread around the world), most participants agreed that COVID-19 remained less severe than EVD because of its seemingly low mortality rate in the country. However, two other participants reported that it had killed more people worldwide than EVD. In addition, some participants referred to community reactions when comparing the two diseases. For example, they claimed that community reactions were more violent during the EVD outbreak because:

‘COVID-19 started at the high level of the worldwide authorities, that is to say: "among the whites" before it happened to us here. It came to our country and it started in the ministries, in the directorates before it came to the neighborhoods until today when we talk about community cases' (IDI 13).

Another HCW added

‘In my opinion, with Ebola it was the fact that people were going to spray the houses. Especially the spraying of the cities, that's what caused enough problems during Ebola. People thought that the spraying was what made the disease worse' (IDI 1).

## Discussion

Our study examined the therapeutic landscape of the Faranah Regional Hospital by exploring the beliefs, practices and experiences of its HCWs' in relation to the COVID-19 pandemic. Our aim was to expand the analytical scope in order to highlight how the pandemic, the preparedness and response plans and the resulting governance consequences were experienced, understood and constructed within the hospital's daily routines. In doing so, we shape the hospital according to the experiences of its most frequent users, turning it into a meaningful social space. This social significance suggests that the perceptions of the HCWs interviewed are a good indicator of their societal values and perspectives on the

pandemic. Their perceptions can also be used to measure the impact on access to healthcare services and the implementation of preparedness and response measures [50]. This is essential for creating a people-centered approach to prevention, preparedness and response, and for identity-based leadership.

### Entanglement with the community

Focusing on some aspects of this therapeutic landscape revealed that our respondents' knowledge was not confined to official and/or scientific information. It also encompassed rumors and non-biomedical treatments employed by the population. Their reliance on different sources to make sense of the pandemic demonstrates that medical practice depends on economic status and social capital [51]. The entanglement of the surveyed HCW's with the communities in which they live and work highlights how porous the boundary between the biomedical world and other social spheres can be. This is evident not only from their knowledge of non-biomedical treatments, but also from the fact that they found explanations for the pandemic in religious worldviews. Similarly, the virus spread and the deployment of response measures were interpreted in line with the country's political situation. This influenced how reliable information sources were perceived (international sources versus national sources).

Research in similar settings has also found a porosity between social and biomedical worlds [37,52,53]. In these studies, hospitals and the social interactions that occur within them are not differentiated by biomedical and non-biomedical practices. Conversely, these practices are considered co-constituent of each other. They occur in multiple spaces (both within and outside the hospital) and overlap the epistemologies of the various actors involved. This is why relationships between actors change over time and shift in configuration as part of syncretic practices. Our findings confirm this multiplicity, so strategies for preparing for and responding to future pandemics must incorporate it, since HCWs appropriated international, national and local health regulations while embodying community perceptions.

### Perceived community differentiation

Another aspect that the therapeutic landscape brought to the fore was that the HCWs interviewed perceived themselves as different from the communities they serve. This differentiation laid in the high value that hospital staff perceived in their knowledge. This led to unease and distrust vis-à-vis patients and community members, particularly when considering the pandemic progression in the country and city. Our respondents described the community as believing rumors and conspiracy theories to explain the origins of the pandemic and using non-biomedical treatments to counteract the disease. Some even argued that the community's level of illiteracy was the underlying reason for these beliefs and non-biomedical practices. In terms of vulnerability perception, the HCWs interviewed for this study also drew a clear distinction between their routines at the hospital and in the city and/or the community. They argued that all protective measures were enforced and complied with in the former, while in the latter there was a relaxation which prevented them from complying and put them in danger.

Psychological and sociological literature has emphasized that pre-existing group identities, norms, values, and worldviews influence people's (non-)compliance with public health advice [27]. This literature has also demonstrated that in-group and out-group memberships influence perceptions of vulnerability during the COVID-19 pandemic [27,54]. Research on HCWs' identity in African settings has stressed that HCWs must constantly negotiate and re-negotiate their identity alongside other seemingly conflicting but potentially complementary social roles [26]. In our study, variations between different HCWs identities (or professions and/or services) did not emerge as a significant point of tension, except with colleagues caring for the few patients who tested positive for SARS-CoV-2 at the hospital. Nonetheless, the distinction was particularly salient between being an HCW (in-group identity) and being a member of the community (out-group). Generally, a strong lack of trust in the healthcare system and social resistance to response teams has been identified during epidemics, but not from HCWs to communities [25,55].

## Experiencing the pandemic was not the same as responding to it

Our results showed that our participants were strongly embedded in the community. In addition, we identified a strong in-group and out-group distinction regarding the community. This pinpoints the discrepancy between what was believed and experienced and what was practiced. Consequently, preparedness and response plans should recognize the elevated risk posed by in-group members, since most people may find it easier to grasp the risk posed by a stranger than by a colleague [56]. This differentiation between perception and practice also appeared to influence healthcare provision during outbreaks and the mental health of the interviewed HCWs, particularly with regard to risk perception and the stress levels when the risk was perceived as high [30]. Likewise, our respondents underestimated the severity of the pandemic, finding the predominance of mild cases reassuring. This has been found to influence the appraisal of the pandemic in several regions of the world other than West Africa, particularly in the early stages of the pandemic [57]. Our findings also underlined how HCWs mobilized, negotiated and translated different practices, discourses and identities to make sense of their lifeworld [41]. Although it may seem contradictory, taking into account the complexity of the beliefs, experiences and practices of HCWs is important for understanding how preparedness and response plans were acted upon locally under particular infrastructural, technical and social pressures [41,58].

## Infodemic management, people-centered approaches and other preparedness and response measures should become operational

The 2021 WHO Strategic Preparedness and Response Plan for the WHO African Region recommended managing the infodemic (an overabundance of information, including misinformation) by ensuring access to evidence-based information from trusted sources at the right time. The Plan also emphasized the safety and security of frontline HCWs in the national preparedness and response plans for COVID-19 [9]. These guidelines stated that African countries would need to move towards people-centered and community-led approaches to the response in order to increase trust and social cohesion. Moreover, lessons learned from the COVID-19 pandemic highlighted community involvement alongside measures to address systemic issues, as well as the management of social resistance and rumors without recourse to violent enforcement [5]. The guidelines also accentuated the need to recruit local staff to build the response structures, including local people (e.g., grassroots and influential community leaders).

**Disinformation and infodemic management.** Our findings confirmed that poor access to quality healthcare services persisted (e.g., a lack of clarity about laboratory results and low reliability of antigen rapid tests). Our results also demonstrated that tensions continued regarding the management of rumors and the official communication of updated information about the pandemic to HCWs and other relevant healthcare system actors. We therefore underline the importance of managing infodemics as an essential tool in preventing, preparing for and responding to outbreaks. Rather than being a top-down mechanism for correcting perceptions, infodemic management should be based on local dynamics. A comprehensive approach is also needed to understand why and how rumors and conspiracy theories arise, and the assumptions on which they are based in terms of their social, political and community significance. The responses from our respondents provide the following evidence for a more context-aware approach to managing infodemics.

Managing infodemics should encompass both social media and also traditional media, such as radio, which is an essential source of information in African societies. In Guinea, research has shown that the radio conveys messages that translate or appropriate global information, while also drawing on the country's political and social history [59]. Trust associated with kinship and power (or hierarchical) relations during face-to-face interactions has also been considered a determining factor in shaping health-related decision-making and behaviors in Guinea [60].The references our interviewees made to social media as an important source of information on the pandemic and of rumors indicate the increasing role that social media is playing on the African continent. However, social media has also been identified as a significant contributor to the spread of misinformation, which threatens society and public health [61]. The relationship between social media and misinformation during epidemics and other public health emergencies still needs to be fully explored in Guinea

and across the African continent. This is particularly important in understanding the concerns and preoccupations of social media users with regard to epidemics. Studying the link between social media and misinformation could also provide insight into how blame is attributed and the potential role of social media in moderating conspiracy theories [62]. Our data indicate a tendency to hold politicians or the country's political situation accountable for the mismanagement of the pandemic alongside the circulation of conspiracy theories that explain this phenomenon.

Recent research on conspiracy theories and pandemics in West Africa has suggested that conspiracy theories should not be interpreted as stable beliefs held by individuals or shared exclusively by certain groups or social circles (e.g. HCWs). Nor are they only prevalent in the margins of societies (e.g., among those with high levels of illiteracy, as mentioned by one of our respondents). Rather, conspiracy theories circulate freely across groups and between individuals who do not necessarily know each other beforehand. Similarly, conspiracy theories convey a wide range of meanings and absorb multiple criticisms, making them an attractive form of discourse [63]. Our data supported this, as HCWs were aware of the conspiracy theories and shared the rumors circulating in their communities. Therefore, infodemics management must consider outbreak preparedness and response interventions, as well as frontline respondents, as sources of rumors and uncertainties. These rumors and uncertainties contain conflicting information regarding infectious and medical risks, and are transmitted to communities at different times during the pandemic. This transmission occurs via formal sectors, such as hospitals and state communiqués, as well as informal sectors, such as markets and social media [64,65].

There is evidence to support the interpretation of the presented data regarding the perception of the COVID-19 against the backdrop of previous outbreak experiences, particularly EVD [54]. This evidence, as well as our own findings, suggests that the correlation between conspiracy theories, the political situation and perceptions of pandemics should be interpreted in terms of social cleavages and divisions. Beyond conveying emotional manipulation for political gain, rumors and conspiracy theories (whether distributed via social media or not) can be understood as reflecting competition among citizens for access to state and corporate resources. Although there is an overabundance of resources during pandemics, these resources are rarely distributed evenly. Our data on the presidential elections should be interpreted in light of these social tensions. These social tensions could also explain the community resistance and the perceptions that the virus originated "among white people" and reached Guinea via ministers who had traveled abroad. Including awareness of these tensions and social cleavages, which may go unnoticed in the daily life, in preparedness and response plans could be useful for operationalizing people-centered approaches and for incorporating social context and trust in infodemic management. Furthermore, as the lessons learned during the EVD response have shown, protocols to correct misinformation should be developed collaboratively with the targeted audiences [66]. Our respondents' awareness of the key people to transmit messages to communities should be enhanced so that infodemic management builds on local logics and concerns, rather than solely seeking to correct rumors or demystify conspiracy theories based on biomedical information.

**Involvement of local people.** In addition, the community reactions encountered by the HCWs and the description of the enforcement of the preparedness measures in the city and at the hospital suggest a low level of active involvement by local people. Our interviewees explained that local leaders and community HCWs did not play a central role in community engagement or in building trust bridges between the healthcare system and the community. It should be noted that the lack of trust and miscommunication between the outbreak response and the community, as described by our respondents, can have historical and colonial roots, exacerbated by every epidemic and period of social unrest. This makes the involvement of local leadership important and necessary, but its role remains contingent on each place and situation [67,68]. Therefore, more efforts are required to incorporate people-centered approaches that account for this contingency. Similarly, further research is needed to understand why such approaches did not become operational despite being highlighted as essential for managing previous epidemics and of other global health emergencies [69–72].

## Strengths and limitations

The study addressed a relevant research gap in the field of health emergency preparedness, prevention and control. It focused on building bridges between people-centered preparedness and response plans, and how these are acted upon by one of the most relevant stakeholders: the HCWs. However, our findings should be analyzed in light of some limitations. The themes discussed in this paper were formulated in the context of a broader study on pandemic preparedness response via a HCWs' Knowledge, Attitudes and Practices (KAP) research. The main results of this study have already been published [73]. As such, our study was exploratory and was not designed to assess how the beliefs, practices and experiences of HCWs change over time as the pandemic evolves. We have published the quantitative KAP results separately from the qualitative results to allow a more detailed presentation of both data sources and to avoid making an overly lengthy mixed-methods article. Furthermore, due to limited time and access to certain individuals or communities, we did not study other groups that are an intrinsic part of the hospital's therapeutic landscape, such as patients, maintenance and cleaning staff, caregivers and families. Similarly, this study did not investigate gender, age, professional experience or ethnicity as variables, and their connection with mistrust and risk perception [34,43,74].

The findings in this article are part of a project, which is a partnership between German and Guinean health authorities aimed at building capacity in IPC, patient safety and quality of care in the Faranah Region. In addition to research, the project provides material and financial support to establish IPC benchmarks at the Faranah Regional Hospital and other primary and secondary healthcare facilities in the Faranah Region. Our research being embedded within this partnership could have constituted a source of information bias. HCWs at the hospital may have been reluctant to provide a comprehensive overview of their views on the pandemic due to the well-known association between the research and the regional health authorities. To minimize this bias, the researcher who collected the data (DM) had previously established a rapport of trust with the HCWs through previous research at the hospital and in the region. MD is a medical doctor trained in Guinea and has professional experience working in the country. Similarly, efforts were made to ensure the representativeness of all professional categories working in the hospital's different wards. We also ensured data reliability through careful supervision of the data collection process by other team members (CR and AM-S), who have experience conducting qualitative and ethnographic research at the hospital, in the region and in the country.

## Conclusion

Our study aimed to deepen understanding of the COVID-19 pandemic in Guinea, starting with healthcare facilities, but also exploring broader processes encompassing social and political phenomena, as well as national and international sanitary regulations. Literature on outbreak preparedness emphasizes epidemiological surveillance through the strengthening of community-led initiatives. We argue that, to effectively incorporate people-centered approaches within these plans, more structural efforts are needed to consider the social landscapes of the disease and improve infodemic management. This includes considering:1) places (such as hospitals) where different social tensions and cleavages related to the outbreak are predominantly salient and 2) individuals (such as HCWs) who are particularly vulnerable, with whom engagement should begin. To prepare for an outbreak in a more comprehensive way, we recommend accounting for the shifting and simultaneous identities of HCWs, as well as the overlapping realities of how pandemics are understood, prioritized, perceived and experienced. This in order to reduce tensions surrounding rumors management and the communication of officially updated information.

## Supporting information

**S1 Table. Data analysis process.**
(DOCX)

**S1 Checklist. Inclusivity in global research.**
(DOCX)

## Acknowledgments

We would like to thank all study participants at the HRF for openly answering our questions and explaining to us their routines and impressions. We would also extend our most sincere appreciation to the Global Health Protection Program (GHPP) of the German Ministry of Health (Bundesministerium für Gesundheit) for funding the project (Grant code: ZMVI1–2519GHP710). Special thanks to Dr. Alpha Oumar Karim Diallo for his collaboration, openness and commitment, and to Dr. Matthias Borchert, Dr. Sophie Müller and Dr. Seth Kofi Abrokwa for critically reviewing earlier versions of the manuscript.

*Reflexivity statement:* This paper originates from a research partnership from the Global Health Protection Program (GHPP) of the German Ministry of Health (Bundesministerium für Gesundheit). It seeks to strengthen co-research and co-writing processes between collaborations between the Global South and the Global North. In the list of authors there is an equal participation of men and women from different academic backgrounds (medical doctors, anthropologists, nurses and global health practitioners) and at different stages of their careers.

## Author contributions

**Conceptualization:** Carlos Rocha, Moussa Douno, Lena Landsmann, Kamis Diallo, Rebekah Wood, Mamadou Diallo.

**Data curation:** Carlos Rocha, Mardjan Arvand.

**Formal analysis:** Carlos Rocha, Moussa Douno, Rebekah Wood, Carolin Meinus, Almudena Marí-Sáez.

**Funding acquisition:** Mamadou Diallo, Almudena Marí-Sáez.

**Investigation:** Carlos Rocha, Moussa Douno, Lena Landsmann, Kamis Diallo, Rebekah Wood, Mardjan Arvand, Carolin Meinus, Mamadou Diallo, Almudena Marí-Sáez.

**Methodology:** Carlos Rocha, Moussa Douno, Kamis Diallo, Rebekah Wood, Almudena Marí-Sáez.

**Project administration:** Carlos Rocha, Lena Landsmann, Kamis Diallo, Rebekah Wood, Mardjan Arvand, Carolin Meinus, Mamadou Diallo.

**Resources:** Carlos Rocha, Mardjan Arvand, Carolin Meinus.

**Supervision:** Carlos Rocha.

**Writing – original draft:** Carlos Rocha, Rebekah Wood, Almudena Marí-Sáez.

**Writing – review & editing:** Moussa Douno, Lena Landsmann, Rebekah Wood, Mardjan Arvand, Carolin Meinus, Mamadou Diallo, Almudena Marí-Sáez.

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
