## [Decision Letter · Decision Letter 0]

12 Aug 2025

Healthcare workers’ beliefs, practices and experiences regarding the COVID-19 pandemic and resulting governance from preparedness and response plans in Faranah, Guinea.

Dear Dr. Rocha,

 The manuscript has been evaluated by two reviewers and their comments are available below.  The reviewers have raised a number of concerns that need attention. They request additional information in the methods section, including further information on study design, ethical issues and the interview guide. They also request further discussion of disinformation and infodemic management and formatting changes to ensure quotations are easy to read.  Could you please revise the manuscript to carefully address the concerns raised?

Kind regards,

Jen Edwards

Staff Editor

Additional Editor Comments (if provided):

Reviewers' comments:

Reviewer's Responses to Questions

**Comments to the Author**

1. Does this manuscript meet PLOS Global Public Health’s publication criteria ? Is the manuscript technically sound, and do the data support the conclusions? The manuscript must describe methodologically and ethically rigorous research with conclusions that are appropriately drawn based on the data presented.

Reviewer #1: Yes

Reviewer #2: Yes

2. Has the statistical analysis been performed appropriately and rigorously?

Reviewer #1: N/A

Reviewer #2: N/A

3. Have the authors made all data underlying the findings in their manuscript fully available (please refer to the Data Availability Statement at the start of the manuscript PDF file)?

Reviewer #1: No

Reviewer #2: No

4. Is the manuscript presented in an intelligible fashion and written in standard English?

Reviewer #1: Yes

Reviewer #2: Yes

Reviewer #1: Manuscript Number: PGPH-D-25-01478

Manuscript Title: Healthcare workers’ beliefs, practices and experiences regarding the COVID-19 pandemic and resulting governance from preparedness and response plans in Faranah, Guinea.

Dear Authors

Thank you for giving me the opportunity to review the paper “Healthcare workers’ beliefs, practices and experiences regarding the COVID-19 pandemic and resulting governance from preparedness and response plans in Faranah, Guinea.”

General Comments

Overall, I think this is a relevant paper. It uses a novel approach to understand the experiences of health workers during the COVID-19 pandemic. I think readers can learn how to conduct social science research during responses to disease outbreaks from the work presented in this paper.

Minor comment: Please change the tense in the entire paper from past participle to past tense since the pandemic was 5 years ago now. From phrases like “the COVID-19 pandemic has highlighted” to “the COVID-19 pandemic highlighted”.

Abstract:

Introduction

Methods:

1. Please state the specific study design for this study. See here: Tenny S, Brannan JM, Brannan GD. Qualitative study. From your methods I see that the specific design was grounded theory

2. Please provide a definition for a health workers. Were janitors and cleaners for example eligible to participate? This cadre work in hospitals and are as likely to contract nosocomial infections just as much as other health workers.

3. Please add the limitations of using convenience sampling as opposed to purposive sampling.

4. Please add a couple of sentences on how the authors ensured trustworthiness. See more here: Elo S, Kääriäinen M, Kanste O, Pölkki T, Utriainen K, Kyngäs H. Qualitative content analysis: A focus on trustworthiness. SAGE open. 2014 Feb 5;4(1):2158244014522633.

5. The data analysis section appears detailed and well described.

6. Please provide more detail about the ethical issues in the study. Was informed consent obtained? Was it verbal or written? How about administrative approval? How about approval from the hospital management?

7. I think the authors should think a little bit deeper about the participant selection. They state that the exclusion criteria were anyone below 18 years or did not provide consent. 1) I don’t think you are going to find a health worker who is below 18 years. 2) Lack of consent cannot be an exclusion criterion because they would not have agreed to participate in the first place. I think there is detail lacking there.

8. Please provide more information about the interview guide. How was it developed? Was it based on a theory or framework? Please cite this.

9. Data collection: Please state that the interviews were held in a language comfortable to the interviewee not the interviewer.

10. I would advise the authors to use the COREQ checklist to ensure all the items are reported for a qualitative study: https://www.equator-network.org/reporting-guidelines/srqr/

Results

11. The presentation of results can be smarter. Please remove the quote from the text in the paragraphs so that they are clearly visible (for example lines 230 – 235). I see that the font size is also different.

12. The authors said that they did the analysis inductively. But from the results I see that the results are presented by category – beliefs, experiences and practices. Please clarify whether the approach was indeed inductive because the results presentation shows that the themes were generated deductively across pre-defined categories of beliefs, experiences and practices.

13. Please move the definition of practice to the methods. Also I would add definitions for the other categories - beliefs and experiences.

Discussion:

14. Could you please add a reflexivity statement in the strengths and limitations. How did the personal biases of the authors influence the findings in this work?

Conclusions

Reviewer #2: This is a useful qualitative analysis for the african healthcare context in public health emmergencies.

"

What is also reassuring for me is when, out of 100 sick

212 people, we say that 80 to 90 have been cured, or out of 100 people, there have only been 2 deaths, that

213 is really reassuring’ (IDI 9). They also perceived information on the development of vaccines as

214 encouraging.

- This should be discussed as underascertainment of mild cases have impacted the severity perception very much in the early pandemic.

"

According to them, rumor

243 sources include "word of mouth", the radio, print and social media"

- Can be futher discussed in light of recent evidence of social media as a main targerted mass desinformation and emotional manipulation initial source ,eventually promoted by opposition parties to capptalize politcally on the pandemic as refered in the next sentence. It will become more important to understand this patterns in global publci health emmergencies.

The authors should duiscuss Infodemic and Infodemics monitoring and management as refered in literature.

"Participants consistently linked the

244 rise of these rumors and conspiracy narratives with the country’s political situation as the pandemic

245 coincided with 2020 presidential elections." -

"4. People-centered approaches and other preparedness and response measures should become

464 operational

- A specific subchapter for desinformation and infodemic management with further discussion i could be useful and facilitate reading. Discussing the role of social media as infodemic source in Guinea and Afriica can be useful. Can it be the origin of rumors even if many people do not have smartphones or low literacy? What is the alphabetization and smartphone use in Guinea?

We formulated the themes discussed in this

500 paper in the context of a broader study on pandemic preparedness response via a HCWs’ Knowledge,

501 Attitudes and Practices research.

- is it published? I wouldn´t mind reading a mixed methods article but understand if it could become too long.

- going through the article it seems that the infodemic detection, antecipation and management is something that should be refered in the abstract and conclusions chapter.

**Do you want your identity to be public for this peer review?** For information about this choice, including consent withdrawal, please see our Privacy Policy .

Reviewer #1: **Yes:** Steven N KabwamaSteven N Kabwama

Reviewer #2: No

---

## [Decision Letter · Decision Letter 1]

26 Nov 2025

PGPH-D-25-01478R1

Healthcare workers’ beliefs, practices and experiences regarding the COVID-19 pandemic and resulting governance from preparedness and response plans in Faranah, Guinea.

Dear Dr. Rocha,

Thank you for submitting your manuscript to PLOS Global Public Health. After careful consideration, we feel that it has merit but does not fully meet PLOS Global Public Health’s publication criteria as it currently stands. Therefore, we invite you to submit a revised version of the manuscript that addresses the points raised during the review process.

The manuscript has been evaluated by two reviewers, and their comments are available below.

The reviewers have raised a number of major concerns. Could you please carefully revise the manuscript to address all comments raised?

We look forward to receiving your revised manuscript.

Kind regards,

Johanna Pruller, Ph.D.

PLOS Staff Editor

Journal Requirements:

Additional Editor Comments (if provided):

Reviewers' comments:

Reviewer's Responses to Questions

**Comments to the Author**

Reviewer #3: All comments have been addressed

Reviewer #4: (No Response)

publication criteria ? Is the manuscript technically sound, and do the data support the conclusions? The manuscript must describe methodologically and ethically rigorous research with conclusions that are appropriately drawn based on the data presented.

Reviewer #3: Yes

Reviewer #4: Yes

3. Has the statistical analysis been performed appropriately and rigorously?

Reviewer #3: Yes

Reviewer #4: Yes

4. Have the authors made all data underlying the findings in their manuscript fully available (please refer to the Data Availability Statement at the start of the manuscript PDF file)?

Reviewer #3: Yes

Reviewer #4: Yes

5. Is the manuscript presented in an intelligible fashion and written in standard English?

Reviewer #3: Yes

Reviewer #4: Yes

Reviewer #3: The manuscript is well-written, methodologically sound, and provides valuable insights into healthcare workers' experiences during the COVID-19 pandemic in Guinea. The revisions have addressed most reviewer concerns, making the paper suitable for publication

Reviewer #4: Dear Authors,

Firstly well done on a wonderful project.

There are some edits and some additional information needed.

The manuscript shows a lot of promise, and is unique in its methodology, its population and the content.

The following are some issues / recommendations

Issues to review regarding your Methods.

- how power dynamics (interviewer known to participants) might bias responses?

- whether recruitment reached saturation across professional role, gender or ward?

No reflection on why only 21 of 74 HCWs participated, and whether non-participants may differ systematically?

It would be nice to show how codes were actually mapped to the framework so please include a schematic or table linking themes to the framework

Beware of Risk of over-generalisation

Several statements imply causal relationships that qualitative interviews cannot support due to the nature of the data being collected. Examples:

“HCWs’ understandings are a good indicator of societal values…” (Discussion) — this is an interpretive leap.

Some claims about “the community” are based only on HCWs' perceptions, not triangulated with community data.

Suggested academic fix: qualify claims consistently with “participants reported/perceived…”.As you can only state what you have which is participants

The flow - Readability

These issues do not undermine scientific quality but affect readability (the flow of the manuscript)

- Several long sentences could be shortened without losing meaning.

- There is inconsistent tense switching between past and present.

- Minor formatting inconsistencies exist (e.g., punctuation in percentages in Table I, typographical inconsistencies around references).

In short well done, but in my opinion you have a little tweeking to bring this manuscript to a higher level.

Best regards

Peer Reviewer

**Do you want your identity to be public for this peer review?** For information about this choice, including consent withdrawal, please see our Privacy Policy .

Reviewer #3: **Yes:** Shazina SaeedShazina Saeed

Reviewer #4: No

---

## [Editor Report · Decision Letter 2]

10 Mar 2026

Healthcare workers’ beliefs, practices and experiences regarding the COVID-19 pandemic and resulting governance from preparedness and response plans in Faranah, Guinea.

PGPH-D-25-01478R2

Dear Scientist Rocha,

We are pleased to inform you that your manuscript 'Healthcare workers’ beliefs, practices and experiences regarding the COVID-19 pandemic and resulting governance from preparedness and response plans in Faranah, Guinea.' has been provisionally accepted for publication in PLOS Global Public Health.

Best regards,

Julia Robinson

Executive Editor